# LEARNING TO ACT BY PREDICTING THE FUTURE

**Alexey Dosovitskiy**
Intel Labs

**Vladlen Koltun**
Intel Labs

## ABSTRACT

We present an approach to sensorimotor control in immersive environments. Our approach utilizes a high-dimensional sensory stream and a lower-dimensional measurement stream. The cotemporal structure of these streams provides a rich supervisory signal, which enables training a sensorimotor control model by interacting with the environment. The model is trained using supervised learning techniques, but without extraneous supervision. It learns to act based on raw sensory input from a complex three-dimensional environment. The presented formulation enables learning without a fixed goal at training time, and pursuing dynamically changing goals at test time. We conduct extensive experiments in three-dimensional simulations based on the classical first-person game Doom. The results demonstrate that the presented approach outperforms sophisticated prior formulations, particularly on challenging tasks. The results also show that trained models successfully generalize across environments and goals. A model trained using the presented approach won the Full Deathmatch track of the Visual Doom AI Competition, which was held in previously unseen environments.

## 1 INTRODUCTION

Machine learning problems are commonly divided into three classes: supervised, unsupervised, and reinforcement learning. In this view, supervised learning is concerned with learning input-output mappings, unsupervised learning aims to find hidden structure in data, and reinforcement learning deals with goal-directed behavior (Murphy, 2012). Reinforcement learning is compelling because it considers the natural setting of an organism acting in its environment. It is generally taken to comprise a class of problems (learning to act), the mathematical formalization of these problems (maximizing the expected discounted return), and a family of algorithmic approaches (optimizing an objective derived from the Bellman equation) (Kaelbling et al., 1996; Sutton & Barto, 2017).

While reinforcement learning (RL) has achieved significant progress (Mnih et al., 2015), key challenges remain. One is sensorimotor control from raw sensory input in complex and dynamic three-dimensional environments, learned directly from experience. Another is the acquisition of general skills that can be flexibly deployed to accomplish a multitude of dynamically specified goals (Lake et al., 2016).

In this work, we propose an approach to sensorimotor control that aims to assist progress towards overcoming these challenges. Our approach departs from the reward-based formalization commonly used in RL. Instead of a monolithic state and a scalar reward, we consider a stream of sensory input $\{\mathbf{s}_t\}$ and a stream of measurements $\{\mathbf{m}_t\}$. The sensory stream is typically high-dimensional and may include the raw visual, auditory, and tactile input. The measurement stream has lower dimensionality and constitutes a set of data that pertain to the agent's current state. In a physical system, measurements can include attitude, supply levels, and structural integrity. In a three-dimensional computer game, they can include health, ammunition levels, and the number of adversaries overcome.

Our guiding observation is that the interlocked temporal structure of the sensory and measurement streams provides a rich supervisory signal. Given present sensory input, measurements, and goal, the agent can be trained to predict the effect of different actions on future measurements. Assuming that the goal can be expressed in terms of future measurements, predicting these provides all the information necessary to support action. This reduces sensorimotor control to supervised learning, while supporting learning from raw experience and without extraneous data. Supervision is pro-

vided by experience itself: by acting and observing the effects of different actions in the context of changing sensory inputs and goals.

This approach has two significant benefits. First, in contrast to an occasional scalar reward assumed in traditional RL, the measurement stream provides rich and temporally dense supervision that can stabilize and accelerate training. While a sparse scalar reward may be the only feedback available in a board game (Tesauro, 1994; Silver et al., 2016), a multidimensional stream of sensations is a more appropriate model for an organism that is learning to function in an immersive environment (Adolph & Berger, 2006).

The second advantage of the presented formulation is that it supports training without a fixed goal and pursuing dynamically specified goals at test time. Assuming that the goal can be expressed in terms of future measurements, the model can be trained to take the goal into account in its prediction of the future. At test time, the agent can predict future measurements given its current sensory input, measurements, and goal, and then simply select the action that best suits its present goal.

We evaluate the presented approach in immersive three-dimensional simulations that require visually navigating a complex three-dimensional environment, recognizing objects, and interacting with dynamic adversaries. We use the classical first-person game Doom, which introduced immersive three-dimensional games to popular culture (Kushner, 2003). The presented approach is given only raw visual input and the statistics shown to the player in the game, such as health and ammunition levels. No human gameplay is used, the model trains on raw experience.

Experimental results demonstrate that the presented approach outperforms state-of-the-art deep RL models, particularly on complex tasks. Experiments further demonstrate that models learned by the presented approach generalize across environments and goals, and that the use of vectorial measurements instead of a scalar reward is beneficial. A model trained with the presented approach won the Full Deathmatch track of the Visual Doom AI Competition, which took place in previously unseen environments. The presented approach outperformed the second best submission, which employed a substantially more complex model and additional supervision during training, by more than 50%.

## 2 BACKGROUND

The supervised learning (SL) perspective on learning to act by interacting with the environment dates back decades. Jordan & Rumelhart (1992) analyze this approach, review early work, and argue that the choice of SL versus RL should be guided by the characteristics of the environment. Their analysis suggests that RL may be more efficient when the environment provides only a sparse scalar reward signal, whereas SL can be advantageous when temporally dense multidimensional feedback is available.

Sutton (1988) analyzed temporal-difference (TD) learning and argued that it is preferable to SL for prediction problems in which the correctness of the prediction is revealed many steps after the prediction is made. Sutton's influential analysis assumes a sparse scalar reward. TD and policy gradient methods have since come to dominate the study of sensorimotor learning (Kober et al., 2013; Mnih et al., 2015; Sutton & Barto, 2017). While the use of SL is natural in imitation learning (LeCun et al., 2005; Ross et al., 2013) or in conjunction with model-based RL (Levine & Koltun, 2013), the formulation of sensorimotor learning from raw experience as supervised learning is rare (Levine et al., 2016). Our work suggests that when the learner is exposed to dense multidimensional sensory feedback, direct future prediction can support effective sensorimotor coordination in complex dynamic environments.

Our approach has similarities to Monte Carlo methods. The convergence of such methods was analyzed early on and they were seen as theoretically advantageous, particularly when function approximators are used (Bertsekas, 1995; Sutton, 1995; Singh & Sutton, 1996). The choice of TD learning over Monte Carlo methods was argued on practical grounds, based on empirical performance on canonical examples (Sutton, 1995). While the understanding of the convergence of both types of methods has since improved (Szepesvári & Littman, 1999; Tsitsiklis, 2002; Even-Dar & Mansour, 2003), the argument for TD versus Monte Carlo is to this day empirical (Sutton & Barto, 2017). Sharp negative examples exist (Bertsekas, 2010). Our work deals with the more general setting of vectorial feedback and parameterized goals, and shows that a simple Monte-Carlo-type method performs extremely well in a compelling instantiation of this setting.

Vector-valued feedback has been considered in the context of multi-objective decision-making (Gábor et al., 1998; Roijers et al., 2013). Transfer across related tasks has been analyzed by Konidaris et al. (2012). Parameterized goals have been studied in the context of continuous motor skills such as throwing darts at a target (da Silva et al., 2012; Kober et al., 2012; Deisenroth et al., 2014). A general framework for sharing value function approximators across both states and goals has been described by Schaul et al. (2015). Our work is most closely related to the framework of Schaul et al. (2015), but presents a specific formulation in which goals are defined in terms of intrinsic measurements and control is based on direct future prediction. We provide an architecture that handles realistic sensory and measurement streams and achieves state-of-the-art performance in complex and dynamic three-dimensional environments.

Learning to act in simulated environments has been the focus of significant attention following the successful application of deep RL to Atari games by Mnih et al. (2015). A number of recent efforts applied related ideas to three-dimensional environments. Lillicrap et al. (2016) considered continuous and high-dimensional action spaces and learned control policies in the TORCS simulator. Mnih et al. (2016) described asynchronous variants of deep RL methods and demonstrated navigation in a three-dimensional labyrinth. Oh et al. (2016) augmented deep Q-networks with external memory and evaluated their performance on a set of tasks in Minecraft. In a recent technical report, Kulkarni et al. (2016b) proposed end-to-end training of successor representations and demonstrated navigation in a Doom-based environment. In another recent report, Blundell et al. (2016) considered a nonparametric approach to control and conducted experiments in a three-dimensional labyrinth. Experiments reported in Section 4 demonstrate that our approach significantly outperforms state-of-the-art deep RL methods.

Prediction of future states in dynamical systems was considered by Littman et al. (2001) and Singh et al. (2003). Predictive representations in the form of generalized value functions were advocated by Sutton et al. (2011). More recently, Oh et al. (2015) learned to predict future frames in Atari games. Prediction of full sensory input in realistic three-dimensional environments remains an open challenge, although significant progress is being made (Mathieu et al., 2016; Finn et al., 2016; Kalchbrenner et al., 2016). Our work considers prediction of future values of meaningful measurements from rich sensory input and shows that such prediction supports effective sensorimotor control.

## 3 MODEL

Consider an agent that interacts with the environment over discrete time steps. At each time step $t$, the agent receives an observation $\mathbf{o}_t$ and executes an action $a_t$ based on this observation. We assume that the observations have the following structure: $\mathbf{o}_t = \langle \mathbf{s}_t, \mathbf{m}_t \rangle$, where $\mathbf{s}_t$ is raw sensory input and $\mathbf{m}_t$ is a set of measurements. In our experiments, $\mathbf{s}_t$ is an image: the agent's view of its three-dimensional environment. More generally, $\mathbf{s}_t$ can include input from multiple sensory modalities. The measurements $\mathbf{m}_t$ can indicate the attitude, supply levels, and structural integrity in a physical system, or health, ammunition, and score in a computer game.

The distinction between sensory input $\mathbf{s}_t$ and measurements $\mathbf{m}_t$ is somewhat artificial: both $\mathbf{s}_t$ and $\mathbf{m}_t$ constitute sensory input in different forms. In our model, the measurement vector $\mathbf{m}_t$ is distinguished from other sensations in two ways. First, the measurement vector is the part of the observation that the agent will aim to predict. At present, predicting full sensory streams is beyond our capabilities (although see the work of Kalchbrenner et al. (2016) and van den Oord et al. (2016) for impressive recent progress). We therefore designate a subset of sensations as measurements that will be predicted. Second, we assume that the agent's goals can be defined in terms of future measurements. Specifically, let $\tau_1, \ldots, \tau_n$ be a set of temporal offsets and let $\mathbf{f} = \langle \mathbf{m}_{t+\tau_1} - \mathbf{m}_t, \ldots, \mathbf{m}_{t+\tau_n} - \mathbf{m}_t \rangle$ be the corresponding differences of future and present measurements. We assume that any goal that the agent will pursue can be defined as maximization of a function $u(\mathbf{f}; \mathbf{g})$. Any parametric function can be used. Our experiments use goals that are expressed as linear combinations of future measurements:

$$u(\mathbf{f}; \mathbf{g}) = \mathbf{g}^\top \mathbf{f}, \tag{1}$$

where the vector $\mathbf{g}$ parameterizes the goal and has the same dimensionality as $\mathbf{f}$. This model generalizes the standard reinforcement learning formulation: the scalar reward signal can be viewed as a measurement, and exponential decay is one possible configuration of the goal vector.

To predict future measurements, we use a parameterized function approximator, denoted by $F$:

$$\mathbf{p}_t^a = F(\mathbf{o}_t, a, \mathbf{g}; \boldsymbol{\theta}). \tag{2}$$

Here $a \in \mathcal{A}$ is an action, $\boldsymbol{\theta}$ are the learned parameters of $F$, and $\mathbf{p}_t^a$ is the resulting prediction. The dimensionality of $\mathbf{p}_t^a$ matches the dimensionality of $\mathbf{f}$ and $\mathbf{g}$. Note that the prediction is a function of the current observation, the considered action, and the goal. At test time, given learned parameters $\boldsymbol{\theta}$, the agent can choose the action that yields the best predicted outcome:

$$a_t = \arg\max_{a \in \mathcal{A}} \mathbf{g}^\top F(\mathbf{o}_t, a, \mathbf{g}; \boldsymbol{\theta}). \tag{3}$$

The goal vector used at test time need not be identical to any goal seen during training.

## 3.1 TRAINING

The predictor $F$ is trained on experiences collected by the agent. Starting with a random policy, the agent begins to interact with its environment. This interaction takes place over episodes that last for a fixed number of time steps or until a terminal event occurs.

Consider a set of experiences collected by the agent, yielding a set $\mathcal{D}$ of training examples: $\mathcal{D} = \{\langle \mathbf{o}_i, a_i, \mathbf{g}_i, \mathbf{f}_i \rangle\}_{i=1}^N$. Here $\langle \mathbf{o}_i, a_i, \mathbf{g}_i \rangle$ is the input and $\mathbf{f}_i$ is the output of example $i$. The predictor is trained using a regression loss:

$$\mathcal{L}(\boldsymbol{\theta}) = \sum_{i=1}^N \left\| F(\mathbf{o}_i, a_i, \mathbf{g}_i; \boldsymbol{\theta}) - \mathbf{f}_i \right\|^2. \tag{4}$$

A classification loss can be used for predicting categorical measurements, but this was not necessary in our experiments.

As the agent collects new experiences, the training set $\mathcal{D}$ and the predictor used by the agent change. We maintain an experience memory of the $M$ most recent experiences out of which a mini-batch of $N$ examples is randomly sampled for every iteration of the solver. The parameters of the predictor used by the agent are updated after every $k$ new experiences. This setup departs from pure on-policy training and we have not observed any adverse effect of using a small experience memory. Additional details are provided in Appendix A.

We have evaluated two training regimes:

1. Single goal: the goal vector is fixed throughout the training process.
2. Randomized goals: the goal vector for each episode is generated at random.

In both regimes, the agent follows an $\varepsilon$-greedy policy: it acts greedily according to the current goal with probability $1 - \varepsilon$, and selects a random action with probability $\varepsilon$. The value of $\varepsilon$ is initially set to 1 and is decreased during training according to a fixed schedule.

## 3.2 ARCHITECTURE

The predictor $F$ is a deep network parameterized by $\boldsymbol{\theta}$. The network architecture we use is shown in Figure 1. The network has three input modules: a perception module $S(\mathbf{s})$, a measurement module $M(\mathbf{m})$ and a goal module $G(\mathbf{g})$. In our experiments, $\mathbf{s}$ is an image and the perception module $S$ is implemented as a convolutional network. The measurement and goal modules are fully-connected networks. The outputs of the three input modules are concatenated, forming the joint input representation used for subsequent processing:

$$\mathbf{j} = J(\mathbf{s}, \mathbf{m}, \mathbf{g}) = \langle S(\mathbf{s}), M(\mathbf{m}), G(\mathbf{g}) \rangle. \tag{5}$$

Future measurements are predicted based on this input representation. The network emits predictions of future measurements for all actions at once. This could be done by a fully-connected module that absorbs the input representation and outputs predictions. However, we found that introducing additional structure into the prediction module enhances its ability to learn the fine differences between the outcomes of different actions. To this end, we build on the ideas of Wang et al. (2016) and

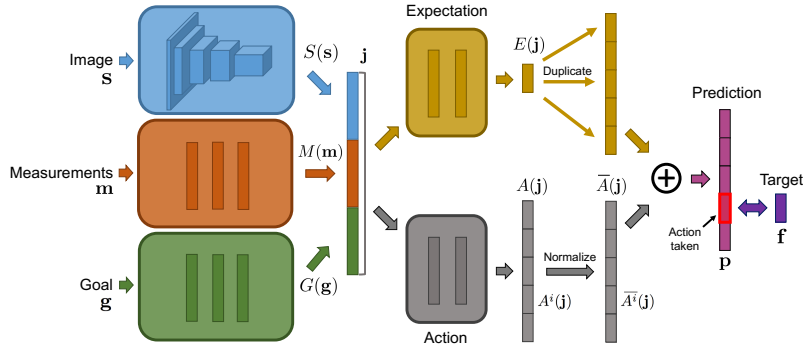

Figure 1: Network structure. The image $\mathbf{s}$, measurements $\mathbf{m}$, and goal $\mathbf{g}$ are first processed separately by three input modules. The outputs of these modules are concatenated into a joint representation $\mathbf{j}$. This joint representation is processed by two parallel streams that predict the expected measurements $E(\mathbf{j})$ and the normalized action-conditional differences $\{\overline{A^i}(\mathbf{j})\}$, which are then combined to produce the final prediction for each action.

split the prediction module into two streams: an expectation stream $E(\mathbf{j})$ and an action stream $A(\mathbf{j})$. The expectation stream predicts the average of the future measurements over all potential actions. The action stream concentrates on the fine differences between actions: $A(\mathbf{j}) = \langle A^1(\mathbf{j}), \ldots, A^w(\mathbf{j}) \rangle$, where $w = |\mathcal{A}|$ is the number of actions. We add a normalization layer at the end of the action stream that ensures that the average of the predictions of the action stream is zero for each future measurement:

$$\overline{A^i}(\mathbf{j}) = A^i(\mathbf{j}) - \frac{1}{w} \sum_{k=1}^{w} A^k(\mathbf{j}) \tag{6}$$

for all $i$. The normalization layer subtracts the average over all actions from each prediction, forcing the expectation stream $E$ to compensate by predicting these average values. The output of the expectation stream has dimensionality $\dim(\mathbf{f})$, where $\mathbf{f}$ is the vector of future measurements. The output of the action stream has dimensionality $w \cdot \dim(\mathbf{f})$.

The output of the network is a prediction of future measurements for each action, composed by summing the output of the expectation stream and the normalized action-conditional output of the action stream:

$$\mathbf{p} = \langle \mathbf{p}^{a_1}, \ldots, \mathbf{p}^{a_w} \rangle = \left\langle \overline{A^1}(\mathbf{j}) + E(\mathbf{j}), \ldots, \overline{A^w}(\mathbf{j}) + E(\mathbf{j}) \right\rangle. \tag{7}$$

The output of the network has the same dimensionality as the output of the action stream.

## 4 EXPERIMENTS

We evaluate the presented approach in immersive three-dimensional simulations based on the classical game Doom. In these simulations, the agent has a first-person view of the environment and must act based on the same visual information that is shown to human players in the game. To interface with the game engine, we use the ViZDoom platform developed by Kempka et al. (2016). One of the advantages of this platform is that it allows running the simulation at thousands of frames per second on a single CPU core, which enables training models on tens of millions of simulation steps in a single day.

We compare the presented approach to state-of-the-art deep RL methods in four scenarios of increasing difficulty, study generalization across environments and goals, and evaluate the importance of different aspects of the model.

### 4.1 SETUP

**Scenarios.** We use four scenarios of increasing difficulty:

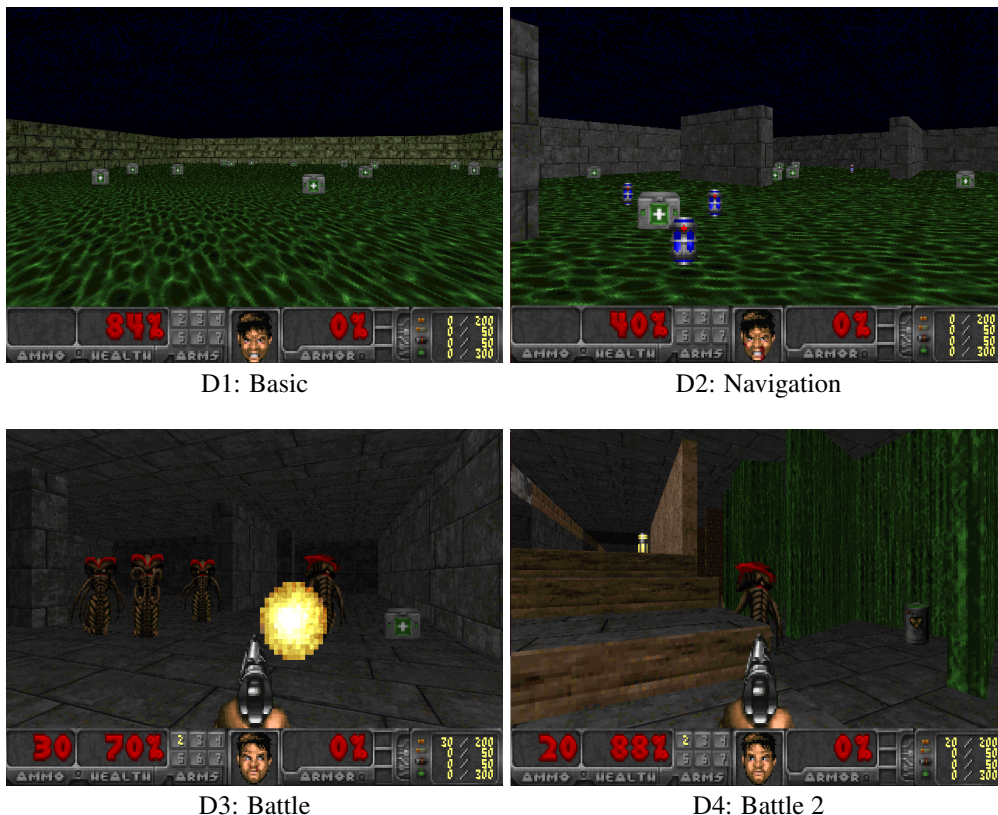

|   |   |
|---|---|
| D1: Basic | D2: Navigation |
| D3: Battle | D4: Battle 2 |

Figure 2: Example frames from the four scenarios.

D1  Gathering health kits in a square room. ("Basic")

D2  Gathering health kits and avoiding poison vials in a maze. ("Navigation")

D3  Defending against adversaries while gathering health and ammunition in a maze. ("Battle")

D4  Defending against adversaries while gathering health and ammunition in a more compli-
    cated maze. ("Battle 2")

These scenarios are illustrated in Figure 2 and in the supplementary video (http://bit.ly/
2f9tacZ).

The first two scenarios are provided with the ViZDoom platform. In D1, the agent is in a square
room and its health is declining at a constant rate. To survive, it must move around and collect health
kits, which are distributed abundantly in the room. This task is easy: as long as the agent learns to
avoid walls and keep traversing the room, performance is good. In D2, the agent is in a maze and
its health is again declining at a constant rate. Here it must again collect health kits that increase its
health, but it must also avoid blue poison vials that decrease health. This task is harder: the agent
must learn to traverse irregularly shaped passageways, and to distinguish health kits from poison
vials. In both tasks, the agent has access to three binary sub-actions: move forward, turn left, and
turn right. Any combination of these three can be used at any given time, resulting in 8 possible
actions. The only measurement provided to the agent in these scenarios is health.

The last two scenarios, D3 and D4, are more challenging and were designed by us using elements of
the ViZDoom platform. Here the agent is armed and is under attack by alien monsters. The monsters
spawn abundantly, move around in the environment, and shoot fireballs at the agent. Health kits and
ammunition are sporadically distributed throughout the environment and can be collected by the
agent. The environment is a simple maze in D3 and a more complex one in D4. In both scenarios,
the agent has access to eight sub-actions: move forward, move backward, turn left, turn right, strafe
left, strafe right, run, and shoot. Any combination of these sub-actions can be used, resulting in

256 possible actions. The agent is provided with three measurements: health, ammunition, and frag count (number of monsters killed).

**Model.**    The future predictor network used in our experiments was configured to be as close as possible to the DQN model of Mnih et al. (2015), to ensure a fair comparison. Additional details on the architecture are provided in Appendix A.

**Training and testing.**    The agent is trained and tested over episodes. Each episode terminates after 525 steps (equivalent to 1 minute of real time) or when the agent's health drops to zero. Statistics reported in figures and tables summarize the final values of respective measurements at the end of episodes.

We set the temporal offsets $\tau_1, \ldots, \tau_n$ of predicted future measurements to 1, 2, 4, 8, 16, and 32 steps in all experiments. Only the latest three time steps contribute to the objective function, with coefficients $(0.5, 0.5, 1)$. More details are provided in Appendix A.

## 4.2    RESULTS

**Comparison to prior work.**    We have compared the presented approach to three deep RL methods: DQN (Mnih et al., 2015), A3C (Mnih et al., 2016), and DSR (Kulkarni et al., 2016b). DQN is a standard baseline for visuomotor control due to its impressive performance on Atari games. A3C is more recent and is commonly regarded as the state of the art in this area. DSR is described in a recent technical report and we included it because the authors also used the ViZDoom platform in experiments, albeit with a simple task. Further details on the setup of the prior approaches are provided in Appendix B.

The performance of the different approaches during training is shown in Figure 3. In reporting the results of these experiments, we refer to our approach as DFP (direct future prediction). For the first two scenarios, all approaches were trained to maximize health. For these scenarios, Figure 3 reports average health at the end of an episode over the course of training. For the last two scenarios, all approaches were trained to maximize a linear combination of the three normalized measurements (ammo, health, and frags) with coefficients $(0.5, 0.5, 1)$. For these scenarios, Figure 3 reports average frags at the end of an episode. Each presented curve averages information from three independent training runs, and each data point is computed from $3 \times 50{,}000$ steps of testing.

DQN, A3C, and DFP were trained for 50 million steps. The training procedure for DSR is much slower and can only process roughly 1 million simulation steps per day. For this reason, we were only able to evaluate DSR on the Basic scenario and were not able to perform extensive hyperparameter tuning. We report results for this technique after 10 days of training. (This time was sufficient to significantly exceed the number of training steps reported in the experiments of Kulkarni et al. (2016b), but not sufficient to approach the number of steps afforded by the other approaches.)

Table 1 reports the performance of the models after training. Each fully trained model was tested over 1 million simulation steps. The table reports average health at the end of an episode for scenarios D1 and D2, and average frags at the end of an episode for D3 and D4. We also report the average training speed for each approach, in millions of simulation steps per day of training. The performance of the different models is additionally illustrated in the supplementary video (`http://bit.ly/2f9tacZ`).

|      | D1 (health)  | D2 (health)  | D3 (frags)   | D4 (frags)    | steps/day |
|------|--------------|--------------|--------------|---------------|-----------|
| DQN  | $89.1 \pm 6.4$   | $25.4 \pm 7.8$   | $1.2 \pm 0.8$    | $0.4 \pm 0.2$     | 7M        |
| A3C  | $\mathbf{97.5 \pm 0.1}$   | $59.3 \pm 2.0$   | $5.6 \pm 0.2$    | $6.7 \pm 2.9$     | 80M       |
| DSR  | $4.6 \pm 0.1$    | —            | —            | —             | 1M        |
| DFP  | $\mathbf{97.7 \pm 0.4}$   | $\mathbf{84.1 \pm 0.6}$   | $\mathbf{33.5 \pm 0.4}$   | $\mathbf{16.5 \pm 1.1}$    | 70M       |

Table 1: Comparison to prior work. We report average health at the end of an episode for scenarios D1 and D2, and average frags at the end of an episode for scenarios D3 and D4.

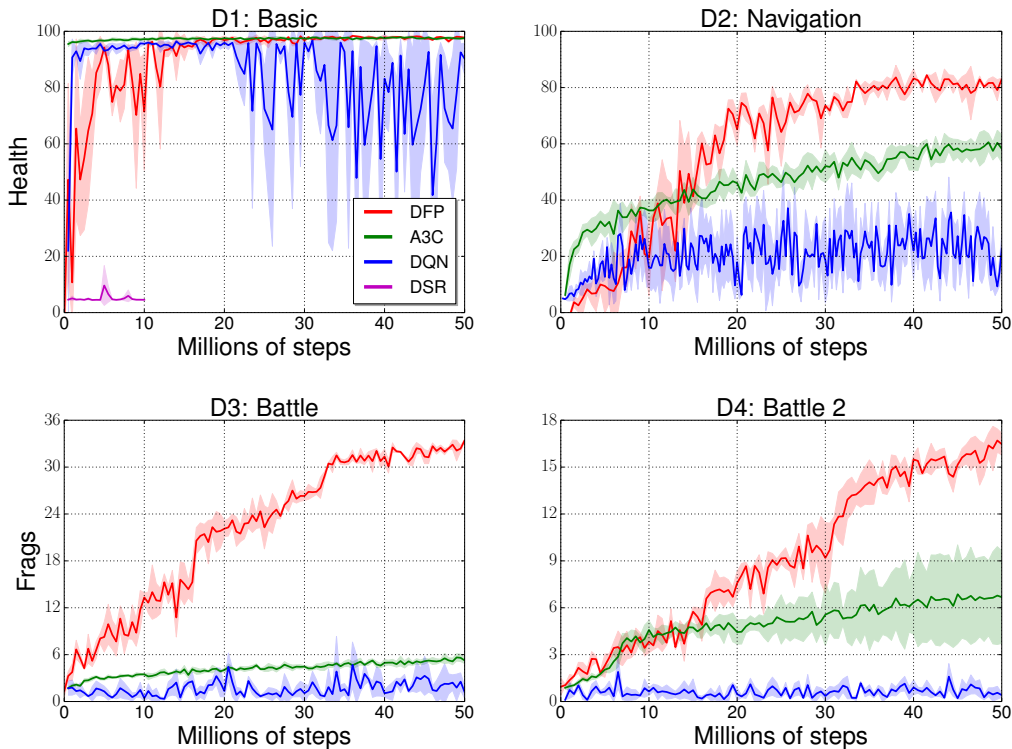

Figure 3: Performance of different approaches during training. DQN, A3C, and DFP achieve similar performance in the Basic scenario. DFP outperforms the prior approaches in the other three scenarios, with a multiplicative gap in performance in the most complex ones (D3 and D4).

In the Basic scenario, DQN, A3C, and DFP all perform well. As reported in Table 1, the performance of A3C and DFP is virtually identical at 97.5%, while DQN reaches 89%. In the more complex Navigation scenario, a significant gap opens up between DQN and A3C; this is consistent with the experiments of Mnih et al. (2016). DFP achieves the best performance in this scenario, with a 25 percentage point advantage during testing. Note that in these first two scenarios, DFP was only given a single measurement per time step (health).

In the more complex Battle and Battle 2 scenarios (D3 and D4), DFP dominates the other approaches. It outperforms A3C at test time by a factor of 6 in D3 and by a factor of 2.5 in D4. Note that the advantage of DFP is particularly significant in the scenarios that provide richer measurements: three measurements per time step in D3 and D4. The effect of multiple measurements is further evaluated in controlled experiments reported below.

**Generalization across environments.** We now evaluate how the behaviors learned by the presented approach generalize across different environments. To this end, we have created 100 randomly textured versions of the mazes from scenarios D3 and D4. We used 90 of these for training and 10 for testing, with disjoint sets of textures in the training and testing environments. We call these scenarios D3-tx and D4-tx.

Table 2 shows the performance of the approach for different combinations of training and testing regimes. For example, the entry in the D4-tx row of the D3 column shows the performance (in average number of frags at the end of an episode) of a model trained in D3 and tested in D4-tx. Not surprisingly, a model trained in the simple D3 environment does not learn sufficient invariance to surface appearance to generalize well to other environments. Training in the more complex multi-texture environment in D4 yields better generalization: the trained model performs well in D3 and exhibits non-trivial performance in D3-tx and D4-tx. Finally, exposing the model to significant variation in surface appearance in D3-tx or D4-tx during training yields very good generalization.

|      |       | Train | | | | |
|------|-------|-------|------|-------|-------|--------|
|      |       | D3    | D4   | D3-tx | D4-tx | D4-tx-L |
| Test | D3    | 33.6  | 17.8 | 29.8  | 20.9  | 22.0   |
|      | D4    | 1.6   | 17.1 | 5.4   | 10.8  | 12.4   |
|      | D3-tx | 3.9   | 8.1  | 22.6  | 15.6  | 19.4   |
|      | D4-tx | 1.7   | 5.1  | 6.2   | 10.2  | 12.7   |

Table 2: Generalization across environments.

The last column of Table 2 additionally reports the performance of a higher-capacity model trained in D4-tx. This combination is referred to as D4-tx-L. As shown in the table, this model performs even better. The architecture is detailed in Appendix A.

**Visual Doom AI Competition.** To further evaluate the presented approach, we participated in the Visual Doom AI Competition, held during September 2016. The competition evaluated sensorimotor control models that act based on raw visual input. The competition had the form of a tournament: the submitted agents play multiple games against each other, their performance measured by aggregate frags. The competition included two tracks. The Limited Deathmatch track was held in a known environment that was given to the participants in advance at training time. The Full Deathmatch track evaluated generalization to previously unseen environments and took place in multiple new environments that were not available to the participating teams at training time. We only enrolled in the Full Deathmatch track. Our model was trained using a variant of the D4-tx-L regime.

Our model won, outperforming the second best submission by more than 50%. That submission, described by Lample & Chaplot (2016), constitutes a strong baseline. It is a deep recurrent Q-network that incorporates an LSTM and was trained using reward shaping and extra supervision from the game engine. Specifically, the authors took advantage of the ability provided by the ViZDoom platform to use the internal configuration of the game, including ground-truth knowledge of the presence of enemies in the field of view, during training. The authors' report shows that this additional supervision improved performance significantly. Our model, which is simpler, achieved even higher performance without such additional supervision.

**Goal-agnostic training.** We now evaluate the ability of the presented approach to learn without a fixed goal at training time, and adapt to varying goals at test time. These experiments are performed in the Battle scenario. We use three training regimes: (a) fixed goal vector during training, (b) random goal vector with each value sampled uniformly from $[0, 1]$ for every episode, and (c) random goal vector with each value sampled uniformly from $[-1, 1]$ for every episode. More details are provided in Appendix A. Intuitively, in the second regime the agent is instructed to maximize the different measurements, but has no knowledge of their relative importance. The third regime makes no assumptions as to whether the measured quantities are desirable or not.

The results are shown in Table 3. Each group of columns corresponds to a training regime and each row to a different test-time goal. Goals are given by the weights of the three measurements (ammo, health, and frags) in the objective function. The first test-time goal in Table 3 is the goal vector used in the battle scenarios in the prior experiments, the second seeks to maximize the frag count, the third is a pacifist (maximize ammo and health, minimize frags), the fourth seeks to aimlessly drain ammunition, and the fifth aims to maximize health. For each row, each group of columns reports the average value of each of the three measurements at the end of an episode. Note that health level at the end of an episode can be negative if the agent suffered major damage in the pre-terminal step.

We draw two main conclusions. First, on the main task (first row), models trained without knowing the goal in advance (b,c) perform nearly as well as a dedicated model trained specifically for the eventual goal (a). Without knowing the eventual goal during training, the agent performs the task almost as well as when it was specifically trained for it. Second, all models generalize to new goals but not equally well. Models trained with a variety of goals (b,c) generalize much better than a model trained with a fixed goal.

| test goal | (a) fixed goal $(0.5, 0.5, 1)$ | | | (b) random goals $[0, 1]$ | | | (c) random goals $[-1, 1]$ | | |
|---|---|---|---|---|---|---|---|---|---|
| | ammo | health | frags | ammo | health | frags | ammo | health | frags |
| $(0.5, 0.5, 1)$ | 83.4 | 97.0 | 33.6 | 92.3 | 96.9 | 31.5 | 49.3 | 94.3 | 28.9 |
| $(0, 0, 1)$ | 0.3 | $-3.7$ | 11.5 | 4.3 | 30.0 | 20.6 | 21.8 | 70.9 | 24.6 |
| $(1, 1, -1)$ | 28.6 | $-2.0$ | 0.0 | 22.1 | 4.4 | 0.2 | 89.4 | 83.6 | 0.0 |
| $(-1, 0, 0)$ | 1.0 | $-8.3$ | 1.7 | 1.9 | $-7.5$ | 1.2 | 0.9 | $-8.6$ | 1.7 |
| $(0, 1, 0)$ | 0.7 | 2.7 | 2.6 | 9.0 | 77.8 | 6.6 | 3.0 | 69.6 | 7.9 |

Table 3: Generalization across goals. Each group of three columns corresponds to a training regime, each row corresponds to a test-time goal. The results in the first row indicate that the approach performs well on the main task even without knowing the goal at training time. The results in the other rows indicate that goal-agnostic training supports generalization across goals at test time.

**Ablation study.** We now perform an ablation study using the D3-tx scenario. Specifically, we evaluate the importance of vectorial feedback versus a scalar reward, and the effect of predicting measurements at multiple temporal offsets. The results are summarized in Table 4. The table reports the performance (in average frags at the end of an episode) of our full model (predicting three measurements at six temporal offsets) and of ablated variants that only predict frags (a scalar reward) and/or only predict at the farthest temporal offset. As the results demonstrate, predicting multiple measurements significantly improves the performance of the learned model, even when it is evaluated by only one of those

|  |  | frags |
|---|---|---|
| all measurements | all offsets | 22.6 |
| all measurements | one offset | 17.2 |
| frags only | all offsets | 10.3 |
| frags only | one offset | 5.0 |

Table 4: Ablation study. Predicting all measurements at all temporal offsets yields the best results.

measurements. Predicting measurements at multiple future times is also beneficial. This supports the intuition that a dense flow of multivariate measurements is a better training signal than a scalar reward.

## 5 DISCUSSION

We presented an approach to sensorimotor control in immersive environments. Our approach is simple and demonstrates that supervised learning techniques can be adapted to learning to act in complex and dynamic three-dimensional environments given raw sensory input and intrinsic measurements. The model trains on raw experience, by interacting with the environment without extraneous supervision. Natural supervision is provided by the cotemporal structure of the sensory and measurement streams. Our experiments have demonstrated that this simple approach outperforms sophisticated deep reinforcement learning formulations on challenging tasks in immersive environments. Experiments have further demonstrated that the use of multivariate measurements provides a significant advantage over conventional scalar rewards and that the trained model can effectively pursue new goals not specified during training.

The presented work can be extended in multiple ways that are important for broadening the range of behaviors that can be learned. First, the presented model is purely reactive: it acts based on the current frame only, with no explicit facilities for memory and no test-time retention of internal representations. Recent work has explored memory-based models (Oh et al., 2016) and integrating such ideas with the presented approach may yield substantial advances. Second, significant progress in behavioral sophistication will likely require temporal abstraction and hierarchical organization of learned skills (Barto & Mahadevan, 2003; Kulkarni et al., 2016a). Third, the presented model was developed for discrete action spaces; applying the presented ideas to continuous actions would be interesting (Lillicrap et al., 2016). Finally, predicting features learned directly from rich sensory input can blur the distinction between sensory and measurement streams (Mathieu et al., 2016; Finn et al., 2016; Kalchbrenner et al., 2016).

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

## A  IMPLEMENTATION DETAILS

### A.1  NETWORK ARCHITECTURES

The detailed architectures of two network variants – `basic` and `large` – are shown in Tables A1 and A2. The `basic` network follows the architecture of Mnih et al. (2015) as closely as possible. The `large` network is similar, but all layers starting from the third are wider by a factor of two. In all networks we use the leaky ReLU nonlinearity $\mathrm{LReLU}(x) = \max(x, 0.2x)$ after each non-terminal layer. We initialize the weights as proposed by He et al. (2015).

| module | input dimension | channels | kernel | stride |
|---|---|---|---|---|
| Perception | $84 \times 84 \times 1$ | 32 | 8 | 4 |
| | $21 \times 21 \times 32$ | 64 | 4 | 2 |
| | $10 \times 10 \times 64$ | 64 | 3 | 1 |
| | $10 \cdot 10 \cdot 64$ | 512 | – | – |
| Measurement | 3 | 128 | – | – |
| | 128 | 128 | – | – |
| | 128 | 128 | – | – |
| Goal | $3 \cdot 6$ | 128 | – | – |
| | 128 | 128 | – | – |
| | 128 | 128 | – | – |
| Expectation | $512 + 128 + 128$ | 512 | – | – |
| | 512 | $3 \cdot 6$ | – | – |
| Action | $512 + 128 + 128$ | 512 | – | – |
| | 512 | $3 \cdot 6 \cdot 256$ | – | – |

Table A1: The `basic` architecture.

| module | input dimension | channels | kernel | stride |
|---|---|---|---|---|
| Perception | $128 \times 128 \times 1$ | 32 | 8 | 4 |
| | $32 \times 32 \times 32$ | 64 | 4 | 2 |
| | $16 \times 16 \times 64$ | 128 | 3 | 1 |
| | $16 \cdot 16 \cdot 128$ | 1024 | – | – |
| Measurement | 3 | 128 | – | – |
| | 128 | 128 | – | – |
| | 128 | 128 | – | – |
| Goal | $3 \cdot 6$ | 128 | – | – |
| | 128 | 128 | – | – |
| | 128 | 128 | – | – |
| Expectation | $1024 + 128 + 128$ | 1024 | – | – |
| | 1024 | $3 \cdot 6$ | – | – |
| Action | $1024 + 128 + 128$ | 1024 | – | – |
| | 1024 | $3 \cdot 6 \cdot 256$ | – | – |

Table A2: The `large` architecture.

We empirically validate the architectural choices in the D3-tx regime. We compare the full `basic` architecture to three variants:

- No normalization: normalization at the end of the action stream is not performed.
- No split: no expectation/action split, simply predict future measurements with a fully-connected network.

- No input measurements: the input measurement stream is removed, and current measurements are not provided to the network.

The results are reported in Table A3. All modifications of the `basic` architecture hurt performance, showing that the two-stream formulation is beneficial and that providing the current measurements to the network increases performance but is not crucial.

|       | full | no normalization | no split | no input measurements |
|-------|------|------------------|----------|-----------------------|
| Score | 22.6 | 21.6             | 16.5     | 19.4                  |

Table A3: Evaluation of different network architectures.

## A.2 OTHER DETAILS

The raw sensory input to the agent is the observed image, in grayscale, without any additional pre-processing. The resolution is 84×84 pixels for the `basic` model and 128×128 pixels for the `large` one. We normalized the measurements by their standard deviations under random exploration. More precisely, we divided ammo count, health level, and frag count by 7.5, 30.0, and 1.0, respectively.

We performed frame skipping during both training and testing. The agent observes the environment and selects an action every $4^{th}$ frame. The selected action is repeated during the skipped frames. This accelerates training without sacrificing accuracy. In the paper, "step" always refers to steps after frame skipping (equivalent to every $4^{th}$ step before frame skipping). When played by a human, Doom runs at 35 frames per second, so one step of the agent is equivalent to 114 milliseconds of real time. Therefore, frame skipping has the added benefit of bringing the reaction time of the agent closer to that of a human.

We set the temporal offsets $\tau_1, \ldots, \tau_n$ of predicted future measurements to 1, 2, 4, 8, 16, and 32 steps in all experiments. The longest temporal offset corresponds to 3.66 seconds of real time. In all experiments, only the latest three predictions (after 8, 16, and 32 steps) contributed to the objective function, with fixed coefficients $(0.5, 0.5, 1.0)$. Therefore, in scenarios with multiple measurements available to the agent (D3 and D4), the goal vector was specified by three numbers: the relative weights of the three measurements (ammo, health, frags) in the objective function. In goal-directed training, these were fixed to $(0.5, 0.5, 1.0)$, and in goal-agnostic training they were sampled uniformly at random from $[0, 1]$ or $[-1, 1]$.

We used an experience memory of $M = 20{,}000$ steps, and sampled a mini-batch of $N = 64$ samples after every $k = 64$ new experiences added. We added the experiences to the memory using 8 copies of the agent running in parallel. The networks in all experiments were trained using the Adam algorithm (Kingma & Ba, 2015) with $\beta_1 = 0.95$, $\beta_2 = 0.999$, and $\varepsilon = 10^{-4}$. The initial learning rate is set to $10^{-4}$ and is gradually decreased during training. The `basic` networks were trained for 800,000 mini-batch iterations (or 51.2 million steps), the `large` one for 2,000,000 iterations.

## B  BASELINES

We compared our approach to three prior methods: DQN (Mnih et al., 2015), DSR (Kulkarni et al., 2016b), and A3C (Mnih et al., 2016). We used the authors' implementations of DQN (`https://github.com/kuz/DeepMind-Atari-Deep-Q-Learner`) and DSR (`https://github.com/Ardavans/DSR`), and an independent implementation of A3C (`https://github.com/muupan/async-rl`). For scenarios D1 and D2 we used the change in health as reward. For D3 and D4 we used a linear combination of changes of the three normalized measurements with the same coefficients as for the presented approach: $(0.5, 0.5, 1)$. For DQN and DSR we tested three learning rates: the default one (0.00025) and two alternatives (0.00005 and 0.00002). Other hyperparameters were left at their default values. For A3C, which trains faster, we performed a search over a set of learning rates ($\{2, 4, 8, 16, 32\} \cdot 10^{-4}$) for the first two tasks; for the last two tasks we trained 20 models with random learning rates sampled log-uniformly between $10^{-4}$ and $10^{-2}$ and random $\beta$ (entropy regularization) sampled log-uniformly between $10^{-4}$ and $10^{-1}$. For all baselines we report the best results we were able to obtain.

