# Peer review of "Learning to Act by Predicting the Future"

_ICLR 2017 — accepted_

[Public Comment · (anonymous) · 06 Dec 2016]
**Meaning for reward,  goal setting**

It's interesting to see that by purely using predictions over the differences between measurements for future step(s) and the presented step, the learning performance could surpass A3C and DSR in Doom. I have the following questions:

(1) In Figure 1, why there is a "Target f" drawn to the right of "Prediction P"? Though this makes the model looks more end-to-end, I guess the "Expectation" component and "Action" component are updating their parameters according to their own regression losses, instead of back-propagating through a unified end. 

(2) The reward formulation in this work is similar to Monte-Carlo methods. In this work, the cumulative reward is represented by summing up the outputs of expectation stream and action stream *over each time steps*. If only those over the last time step is considered, it will become almost same as Monte-Carlo. However, the presented work accumulates the difference over multiple future steps. Based on the proposed form of *f* (in Section 3), the reward collected by near-future steps are weighted more. Or we could see the weight to be interpreted as linearly decaying based on the time distance from the future step to the current step. If this is the case, re-formulating the proposed approach with linearly discounted Monte-Carlo could be possible. You may also compare such Monte-Carlo as a compelling baseline. Also, it's better to put more discussion over your reward formulation, since it contributes a lot to the performance improvement and it's better to let people understand why it works well. 

(3) I don't quite understand what is the meaning of *goal*. In Section 3, *g* is proposed to be of the same dimension as  *f*. The maximization is over g^T*f, but what the architecture try to regress is over f, and g is designed as part of inputs to the deep neural network. And action-selection is over the maximum of summed E and A over all future steps. In this way, what the agent is trying to do is to maximize the future gain over those measurements, instead of performing according to the specified goal. Hope the authors could clarify on this.

[Official Review · AnonReviewer3 · rating 8 · confidence 4 · 16 Dec 2016]
**No Title**
soundness 4 · originality 4 · substance 2 · meaningful comparison 2

This paper presents an on-policy deep RL method with additional auxiliary intrinsic variables. 

- The method is a special case of an universal value function based approach and the authors do cite the correct references. Maybe the biggest claimed technical contribution of this paper is to distill many of the existing ideas to solve 3D navigation problems. I think the contributions should be more clearly stated in the abstract/intro

- I would have liked to see failure modes of this approach. Under what circumstances does the model have problems generalizing to changing goals? There are other conceptual problems -- since this is an on-policy method, there will be catastrophic forgetting if the agent dosen't repeatedly train on goals from the distant past. 

- Since the main contribution of this paper is to integrate several key ideas and show empirical advantage, I would have liked to see results on other domains like Atari (maybe using the ROM as intrinsic variables)

Overall, I think this paper does show clear empirical advantage of using the proposed underlying formulations and experimental insights from this paper might be valuable for future agents

[Official Review · AnonReviewer1 · rating 8 · confidence 4 · 16 Dec 2016]
soundness 3 · originality 5 · clarity 4 · impact 3 · substance 3

The paper presents an on-policy method to predict future intrinsic measurements. All the experiments are performed in the game of Doom (vizDoom to be exact), and instead of just predicting win/loss or the number of frags (score), the authors trained their model to predict (a sequence of) triplets of (health, ammunition, frags), weighted by (a sequence of) "goal" triplets that they provided as input. Changing the weights of the goal triplet is a way to perform/guide exploration. At test time, one can act by maximizing the long term goal only.

The results are impressive, as this model won the 2016 vizDoom competition. The experimental section of the paper seems sound:
 - There are comparisons of DFP with A3C, DQN, and an attempt to compare with DSR (a recent similar approach from Kulkarni et al., 2016). DFP outperforms other approaches (or equal them when they reach a ceiling / optimum, as for A3C in scenario D1).
 - There is an ablation study that supports the thesis that all the "added complexity" of the paper's model is useful.

Predicting intrinsic motivation (Singh et al. 2004), auxiliary variables, and forward modelling, are well-studied domains of reinforcement learning. The version that I read (December 4th revision) adequately references prior work, even if it is not completely exhaustive. 

A few comments (nitpicks) on the form:
 - Doom is described as a 3D environment, whereas it is actually a 2D environment (the height is not a discriminative/actionable dimension) presented in (fake) 3D.
 - The use of "P" in (2) (and subsequently) may be misleading as it stands for prediction but not probability (as is normally the case for P).
 - The double use of "j" (admittedly, with different fonts) in (6) may be misleading.
 - Results tables could repeat the units of the measurements (in particular as they are heterogenous in Table 1).

I think that this paper is a clear accept. One could argue that experiments could be conducted on different environments or that the novelty is limited, but I feel that "correct" (no-nonsense, experimentally sound on Doom, appendix providing details for reproducibility) and "milestone" (vizDoom winner) papers should get published.

[Official Review · AnonReviewer2 · rating 7 · confidence 4 · 17 Dec 2016]
**Compelling empirically driven result**
originality 4 · clarity 5 · recommendation (unofficial) 2

Deep RL (using deep neural networks for function approximators in RL algorithms) have had a number of successes solving RL in large state spaces. This empirically driven work builds on these approaches. It introduces a new algorithm which performs better in novel 3D environments from raw sensory data and allows better generalization across goals and environments. Notably, this algorithm was the winner of the Visual Doom AI competition.

The key idea of their algorithm is to use additional low-dimensional observations (such as ammo or health which is provided by the game engine) as a supervised target for prediction. Importantly, this prediction is conditioned on a goal vector (which is given, not learned) and the current action. Once trained the optimal action for the current state can be chosen as the action that maximises the predicted outcome according the goal. Unlike in successor feature representations, learning is supervised and there is no TD relationship between the predictions of the current state and the next state.

There have been a number of prior works both in predicting future states as part of RL and goal driven function approximators which the authors review in section 2. The key contributions of this work are the focus on Monte Carlo estimation (rather than TD), the use of low-dimensional ‘measurements’ for prediction, the parametrized goals and, perhaps most importantly, the empirical comparison to relevant prior work.

In addition to the comparison with Visual Doom AI, the authors show that their algorithm is able to learn generalizable policies which can respond, without further training, to limited changes in the goal.

The paper is well-communicated and the empirical results compelling and will be of significant interest.

Some minor potential improvements:
There is an approximation in the supervised training as it is making an on-policy assumption but it learns from a replay buffer (with the Monte Carlo regression the expectation of the remainder of the trajectory is assumed to follow the current policy, but is being sampled from episodes generated by prior versions of the policy). This should be discussed.
The algorithm uses additional metadata (the information about which parts of the sensory input are worth predicting) that the compared algorithms do not. I think this, and the limitations of this approach (e.g. it may not work well in a sensory environment if such measurements are not provided) should be mentioned more clearly.

[Author Response · Alexey Dosovitskiy · 12 Jan 2017]
**Response to reviewers**

We thank the reviewers for their work and their comments. The reviews will help in further improving the paper. Some specific responses to individual reviewers are below.


AnonReviewer1:

Thank you, we will further polish up the notation and the tables based on your suggestions.


AnonReviewer2:

We used a very small replay buffer (2,500 frames per actor thread, 20,000 frames in total), and have not observed any significant changes in results when making it smaller or larger. This indicates that in practice the algorithm is not constrained to strict on-policy training. We will discuss this in more detail in the paper.

We agree that measurements that can be used for future prediction are not always available. (For example, as mentioned in the introduction, board games are an extreme example in which only a sparse scalar reward is provided.) We will further emphasize this in the paper to remind the reader that the presented approach adopts different modeling assumptions from most approaches in the literature.


AnonReviewer3:

The simplicity of our model leads to several failure modes: the agent acts purely reactively, cannot construct a persistent representation of its environment (e.g., a cognitive map), and cannot plan its actions far into the future. For this reason its performance in Doom is at the level of a very inexperienced human player. Much exciting work remains to be done.

We are interested in sensorimotor control in immersive environments and chose Doom for this reason. We are looking forward to testing and further developing the approach in other simulators that are becoming available, such as DeepMind's new Lab.

[Final Decision · Program Chairs · 06 Feb 2017]
**ICLR committee final decision**

This paper details the approach that won the VizDoom competition - an on-policy reinforcement learning approach that predicts auxiliary variables, uses intrinsic motivation, and is a special case of a universal value function. The approach is a collection of different methods, but it yields impressive empirical results, and it is a clear, well-written paper.